# Non-Stationary Bandits under Recharging Payoffs: Improved Planning with Sublinear Regret

**Orestis Papadigenopoulos**
Department of Computer Science
The University of Texas at Austin
papadig@cs.utexas.edu

**Constantine Caramanis**
Department of Electrical and Computer Engineering
The University of Texas at Austin
constantine@utexas.edu

**Sanjay Shakkottai**
Department of Electrical and Computer Engineering
The University of Texas at Austin
sanjay.shakkottai@utexas.edu

## Abstract

The stochastic multi-armed bandit setting has been recently studied in the non-stationary regime, where the mean payoff of each action is a non-decreasing function of the number of rounds passed since it was last played. This model captures natural behavioral aspects of the users which crucially determine the performance of recommendation platforms, ad placement systems, and more. Even assuming prior knowledge of the mean payoff functions, computing an optimal planning in the above model is NP-hard, while the state-of-the-art is a $1/4$-approximation algorithm for the case where at most one arm can be played per round. We first focus on the setting where the mean payoff functions are known. In this setting, we significantly improve the best-known guarantees for the planning problem by developing a polynomial-time $(1 - 1/e)$-approximation algorithm (asymptotically and in expectation), based on a novel combination of randomized LP rounding and a time-correlated (interleaved) scheduling method. Furthermore, our algorithm achieves improved guarantees – compared to prior work – for the case where more than one arm can be played at each round. Moving to the bandit setting, when the mean payoff functions are initially unknown, we show how our algorithm can be transformed into a bandit algorithm with sublinear regret.

## 1   Introduction

In the last two decades, the predominant rise of the social media industry has made the notion of a *newsfeed* an integral part of our lives. In a newsfeed, a user observes a structured sequence of content items (posts, photos etc.) particularly selected by the platform according to her/his preferences. Apart from social media, an analogous idea – potentially relabeled – also appears in different domains as, for example, "frequently bought together" in e-commerce, "shuffling similar songs" in music recommendation, or "recommended articles" in scholarly literature indexing databases. Whether it is measured in terms of click-rate or time devoted, the high-level objective of newsfeeds is fairly well-known: to maximize the user's engagement with the platform. In many applications, however, achieving this objective is not as simple as identifying the user's "favorite" content, given that her/his satisfaction can depend on the time passed since the same (or similar) content has been observed. As an example, a user's engagement can worsen if a social media feed (resp., a music recommendation platform) constantly presents content from the same source (resp., same artist).

36th Conference on Neural Information Processing Systems (NeurIPS 2022).

Motivated by such scenarios, researchers have recently studied online decision making problems capturing the notion of "recovering" payoffs, namely, scenarios where the payoff of an action drops (to zero) after each play and then slowly increases back to a baseline. In the context of online learning, these non-stationary models interpolate between multi-armed bandits, where the environment is assumed to be intact, and reinforcement learning, since the actions may now alter the future environment in a structured manner. These models are wide enough to capture many real-life applications, yet special enough to accept efficiently computable (near-)optimal solutions. In this direction, the following general model was first introduced (under slightly different assumptions) by Immorlica and Kleinberg [IK18] and has been recently studied by Simchi-Levi et al. [SLZZ21]:

**Problem** (Multi-armed bandits under recharging payoffs). *We consider a set of* actions *(or* arms*), where each arm $i$ is associated with a* mean payoff function $p_i$. *For each $\tau$, $p_i(\tau)$ is the expected payoff collected for playing action $i$, when $i$ has been played before exactly $\tau$ rounds (we refer to $\tau$ as the* delay *of an arm at a specific round). For each arm, we assume that its payoff function: (a) is monotone non-decreasing in $\tau$, and (b) has a known finite* recovery time*, namely, a specific delay after which the function remains constant. At each round, the player pulls a subset of at most $k$ actions, observes the realized payoffs of the arms played (each computed with respect to the delay of the corresponding action), and collects their sum. The objective is to maximize the expected cumulative payoff collected within an unknown time horizon.*

We denote by $k$-RB an instance of the problem where at most $k$ arms can be played at each round. Further, we distinguish between the *planning* setting, where the payoff functions are known to the player a priori (thus, payoffs can be considered deterministic), and the *learning* setting, where these are initially unknown and the user assumes semi-bandit feedback on the payoffs of the played actions. In the former case, the goal is to design an efficient algorithm which closely-approximates the optimal planning (since this is generally an NP-hard problem), while in the latter, the objective is to construct a bandit algorithm of sublinear regret, defined as the difference in expected payoff between a planning algorithm and its bandit variant due to the initial absence of information of the latter.

## 1.1 Background and Related Work

Immorlica and Kleinberg [IK18] first study the 1-RB problem and provide a $(1 - \epsilon)$-approximation (asymptotically) for the planning setting, under the additional assumption that the payoff functions are weakly concave over the whole time horizon. On the other extreme, Basu et al. [BSSS19] provide a long-run $(1 - 1/e)$-approximation for the case where, after each play, an arm *cannot* be played again for a fixed number of rounds – a problem which can be cast as a special instance of 1-RB using Heaviside step payoff functions.[1] Shortly after the introduction of the problem, a number of special cases and variations have been studied [CCB20, PC21, PBG19, BPCS21, BCMTT20, CCB20, LCCBGB22, APB⁺21] (we address the reader to Appendix A for an overview).

In their recent work, Simchi-Levi et al. [SLZZ21] study the $k$-RB problem and prove the first $\mathcal{O}(1)$-approximation guarantee under no additional assumptions. For large $k$, this result is the state-of-the-art. However, for small $k$, understanding the magnitude of the constant becomes the primary theoretical question. Specifically, the motivating case of newsfeeds (when, for instance, the content items are presented to a user sequentially) can be modeled as an instance of 1-RB. In this case, the approximation guarantee of [SLZZ21] becomes $1/4$, which is significantly weaker compared to both the $(1 - \epsilon)$-approximation of [IK18] for concave functions and the $(1 - 1/e)$-approximation of [BSSS19] for the most "extreme" example of monotone convex functions, i.e., that of Heaviside step functions. The above observation indicates that either the approximability status of the problem is not well-understood, or that the problem does not gradually become "easier" by increasing the concavity of the payoff functions.

## 1.2 Technical Contributions

In our work, we resolve this discrepancy by designing a long-run $(1 - 1/e)$-approximation algorithm (in expectation) for 1-RB, which improves the state-of-the-art due to [SLZZ21]. Simultaneously, our algorithm enjoys the same asymptotic guarantee of $(1 - \mathcal{O}(1/\sqrt{k}))$ for the general case of $k$-RB as in Simchi-Levi et al. [SLZZ21] with improved and explicit constants, as opposed to the guarantees of [SLZZ21] which do not come in a closed-form (see Table I for a summary of the best-known

---

[1] Recall that the Heaviside step function is defined such that $H(\tau) = 1$ for $\tau \geq 0$, and $H(\tau) = 0$, otherwise.

| Payoff functions | State-of-the-art | Reference | This work |
|---|---|---|---|
| Non-decreasing, concave, Lipschitz | $(1 - \epsilon)$-PTAS for $k = 1$ | [IK18] | – |
| Scaled/translated Heaviside step | $1 - \frac{1}{e}$ for $k = 1$ | [BSSS19, PC21] | – |
| // | $\max_{a \in \mathbb{Z}_+} \frac{a}{a+1} \frac{k}{k+a}$ for any $k$ | [SLZZ21] | $1 - \frac{k^k}{e^k k!}$ |
| Non-decreasing (finite recovery) | $\frac{1}{4}$ for $k = 1$ | [SLZZ21] | $1 - \frac{1}{e}$ |
| // | $\max_{a \in \mathbb{Z}_+} \frac{a}{a+1} \frac{k}{k+a}$ for any $k$ | [SLZZ21] | $1 - \frac{k^k}{e^k k!}$ |

Table 1: Summary of existing results (approximation guarantees) and our improvements.

approximation guarantees). Our algorithm is based on a novel combination of linear programming (LP) rounding and a time-correlated (interleaved) scheduling method, and is significantly simpler to implement compared to prior work. For the case where the mean payoff functions are initially unknown, we show how our algorithm can be transformed into a bandit algorithm with sublinear regret guarantees.

### 1.3 Key Ideas and Intuition

Henceforth, we refer to any instance where the payoff function of each action $i$ has the form $q_i(\tau) = p_i \cdot H(\tau - d_i)$, where $p_i \in [0, 1]$, $d_i \in \mathbb{N}$, and $H(\cdot)$ the Heaviside step function, as the *Heaviside k-RB* problem. The fact that our $(1 - 1/e)$-approximation for 1-RB matches the state-of-the-art bound obtained for Heaviside 1-RB [BSSS19, PC21] is no coincidence. Our solution technique shows that each $k$-RB problem is in fact (approximately) hiding a Heaviside $k$-RB problem.

**Structural characterization of a natural LP relaxation.** A key idea in both [IK18] and [SLZZ21] is to construct a concave relaxation of the optimal solution. Instead, we take a more direct approach and construct a natural LP relaxation (see Section 3) of the optimal average payoff (our construction uses a uniform upper bound on the recovery time of any arm). By carefully analyzing the structure of our LP, we prove that its extreme point solutions follow a particularly interesting *sparsity pattern*: there exist unique delays $\{\tau_i\}$ associated with the arms (which we refer to as "critical" delays), such that playing each arm $i$ exactly once every $\tau_i$ rounds matches the average payoff of the relaxation. As we show, an exception to the above rule can be at most a single arm, which we refer to as *irregular*. The above observation already hints that our problem could potentially be reduced to an instance of Heaviside $k$-RB for which better approximation guarantees (compared to [SLZZ21]) are known to exist (at least for small $k$) [BSSS19, PC21].

**Improved approximation guarantees for the planning $k$-RB.** Constructing a planning where each arm is played at a rate indicated by its critical delay (as described above) is generally infeasible, since these rates stem from a relaxation of the problem. However, targeting these rates becomes the starting point of our approximation. In Section 4, we design a novel algorithm for the $k$-RB problem, called "Randomize-Then-Interleave", which starts from computing an optimal solution to our LP relaxation. By combining the sparse structure of this solution with a *randomized rounding* step for determining a critical delay for the (possible) irregular arm, our algorithm produces a "proxy" instance of the Heaviside $k$-RB problem. Finally, the algorithm applies the *interleaved scheduling* method of [PC21] on the resulting Heaviside $k$-RB instance. As we prove, the above algorithm achieves a long-run $(1 - \mathcal{O}(1/\sqrt{k}))$-approximation guarantee (in expectation), but with improved and explicit constants compared to [SLZZ21]. Importantly, for the particular case of $k$-RB, our guarantee is stronger than the one provided in the original work of [PC21].

As a first step, in Section 4.1 we study the performance of our algorithm under the simplifying assumption that the solution returned by our LP relaxation does not contain an irregular arm, that is, every arm is associated with a unique critical delay. For this case, the approximation guarantee of our algorithm follows by critically leveraging existing results on the *correlation gap* of the weighted rank function of uniform matroids due to [Yan11] (see Section 2). In Section 4.2, we relax this simplifying assumption by carefully studying the contribution of the irregular arm to the produced solution. Our strategy relies on the novel idea of considering parallel (fictitious) copies of the irregular arm, one for each possible critical delay. While the marginal probabilities of each copy being played under the corresponding delay at any round are consistent with the initial LP solution, these events are *mutually*

*exclusive*. Nevertheless, we show that expected payoff collected can only decrease by assuming that the parallel copies were instead independent. Having established that, the analysis reduces back to the "easier" case where an irregular arm does not exist.

**A bandit adaptation with sublinear regret.**   Finally, we turn our focus to the learning setting, where the mean payoff functions are initially unknown, and the player has semi-bandit feedback on the realized payoffs of the played arms of each round. In this setting, we begin by analyzing the robustness of our planning algorithm under small perturbations of the payoff functions. Then, we provide sample complexity results to bound the number of samples required to get accurate estimates of the mean payoff functions. By combining the above elements, we transform our planning algorithm into a bandit one, based on an Explore-then-Commit (ETC) scheme, and prove that the latter achieves sublinear regret relative to its planning counterpart. We present these results in Section 5.

All the omitted proofs of our results have been moved to the Appendix.

## 2   Preliminaries

**Problem definition and notation.**   We consider a set $\mathcal{A} = \{1, 2, \ldots, n\}$ of $n$ *actions* (or *arms*), where each arm $i \in \mathcal{A}$ is associated with a (mean) *payoff function* $p_i : \mathbb{N} \to [0, 1]$. For each $\tau \in \mathbb{N}$, $p_i(\tau)$ is the expected payoff for playing action $i \in \mathcal{A}$, when $i$ has been played before exactly $\tau$ rounds (i.e., $\tau = 1$ if the action has been played in the previous round). We refer to the number of rounds since the last play of an arm as the *delay*. For each arm $i \in \mathcal{A}$, we assume that the function $p_i(\tau)$: (a) is monotone non-decreasing in $\tau$, and (b) has a polynomial *recovery time* $\tau_i^{\max}$ such that $p_i(\tau) = p_i(\tau_i^{\max})$ for every $\tau \geq \tau_i^{\max}$. We assume knowledge of a universal upper bound $\tau^{\max}$ on the recovery time of all arms, i.e., $\tau^{\max} \geq \max_{i \in \mathcal{A}} \tau_i^{\max}$.[2] At each round, the player plays a subset of at most $k < n$ actions and collects the sum of their associated payoffs (each computed with respect to the delay of the corresponding action at the time it is played). The objective is to maximize the cumulative payoff collected within an (unknown) horizon of $T$ rounds. We refer to the above instance as $k$-RB.

As a convention, we assume that the delay of each arm $i$ is initially equal to 1 (i.e., all arms are played at time $t = 0$). Note that, even if we assume that all arms start with delay $\tau^{\max}$ (i.e., the arms have never been played before), our results do not change qualitatively.

For any non-negative integer $n \in \mathbb{N}$, we define $[n] = \{1, 2, \ldots, n\}$. For any vector $\mathbf{w} \in \mathbb{R}^n$ and set $S \subseteq [n]$, we denote $\mathbf{w}(S) = \sum_{i \in S} w_i$. Finally, throughout this work we assume that, when comparing between different payoffs, ties are broken arbitrarily.

**Continuous extensions and the correlation gap.**   Let $f : 2^{[n]} \to [0, \infty)$ be a set function defined over a finite set of $n$ elements. For any vector $\mathbf{y} \in [0, 1]^n$, we denote by $\mathcal{D}(\mathbf{y})$ a distribution over $2^{[n]}$ with marginal probabilities $\mathbf{y} = (y_i)_{i \in [n]}$. We denote by $S \sim \mathcal{D}(\mathbf{y})$ a random subset $S \subseteq [n]$ sampled from the distribution $\mathcal{D}(\mathbf{y})$. Further, we denote by $S \sim \mathcal{I}(\mathbf{y})$ a random subset $S \subseteq [n]$ which is constructed by adding each element $i \in [n]$ to $S$, independently, with probability $y_i$.

We recall two canonical continuous extensions of a set function (see [CCPV07, Sch03]):

**Definition 2.1** (Multi-linear extension). *For any vector $\mathbf{y} \in [0, 1]^n$, the* multi-linear extension *of a set function $f$ is defined as*

$$F(\mathbf{y}) = \mathop{\mathbb{E}}_{S \sim \mathcal{I}(\mathbf{y})} [f(S)] = \sum_{S \subseteq [n]} f(S) \prod_{i \in S} y_i \prod_{i \notin S} (1 - y_i).$$

**Definition 2.2** (Concave closure). *For any vector $\mathbf{y} \in [0, 1]^n$, the* concave closure *of a set function $f$ is defined as*

$$f^+(\mathbf{y}) = \sup_{\mathcal{D}(\mathbf{y})} \mathop{\mathbb{E}}_{S \sim \mathcal{D}(\mathbf{y})} [f(S)] = \sup_{\boldsymbol{\alpha}} \left\{ \sum_{S \subseteq [n]} \alpha_S f(S) \mid \sum_{S \subseteq [n]} \alpha_S \mathbf{1}_S = \mathbf{y}, \sum_{S \subseteq [n]} \alpha_S = 1, \boldsymbol{\alpha} \succeq 0 \right\},$$

*where $\mathbf{1}_S \in \{0, 1\}^n$ is an indicator vector such that $(\mathbf{1}_S)_i = 1$, if $i \in S$, and $(\mathbf{1}_S)_i = 0$, otherwise.*

---

[2]Due to the polynomial recovery time assumption, oracle access to the payoff functions is not required.

For any non-negative weight vector $\mathbf{w} \in [0, \infty)^n$, of particular importance in the analysis of our algorithm is the function $f_{\mathbf{w},k} : 2^{[n]} \to [0, \infty)$, defined as

$$f_{\mathbf{w},k}(S) = \max\{\mathbf{w}(I) \mid I \subseteq S \text{ and } |I| \leq k\}. \tag{1}$$

We remark that $f_{\mathbf{w},k}$ corresponds to the *weighted rank function* of the (rank-$k$) uniform matroid under a weight vector $\mathbf{w}$ and, hence, is non-decreasing submodular [Sch03].[3] We denote by $F_{\mathbf{w},k}$ and $f_{\mathbf{w},k}^+$ the multi-linear extension and the concave closure of $f_{\mathbf{w},k}$, respectively.

The *correlation gap* [ADSY10] of a set function is defined as $\sup_{\mathbf{y} \in [0,1]^n} \frac{f^+(\mathbf{y})}{F(\mathbf{y})}$. The following result due to [Yan11] provides an upper bound on the correlation gap of $f_{\mathbf{w},k}(\cdot)$:

**Lemma 2.3** (Correlation gap [Yan11])**.** *Let $f_{\mathbf{w},k} : 2^{[n]} \to [0, \infty)$ be the* weighted rank function *of the rank-$k$ uniform matroid. Then, for any vector $\mathbf{y} \in [0,1]^n$ we have*

$$f_{\mathbf{w},k}^+(\mathbf{y}) \geq F_{\mathbf{w},k}(\mathbf{y}) \geq \left(1 - \frac{k^k}{e^k k!}\right) \cdot f_{\mathbf{w},k}^+(\mathbf{y}) \approx \left(1 - \frac{1}{\sqrt{2\pi k}}\right) \cdot f_{\mathbf{w},k}^+(\mathbf{y}).$$

# 3 Structural Properties of a Natural LP Relaxation

We consider the following LP relaxation which applies to any planning instance of $k$-RB:

$$\underset{\mathbf{x} \succeq \mathbf{0}}{\textbf{maximize:}} \quad \sum_{i \in \mathcal{A}} \sum_{\tau \in \mathbb{N}} p_i(\tau) \cdot x_{i,\tau} \tag{$\mathbf{LP}_k$}$$

$$\textbf{s.t.:} \quad \sum_{i \in \mathcal{A}} \sum_{\tau \in \mathbb{N}} x_{i,\tau} \leq k, \tag{C.1}$$

$$\sum_{\tau \in \mathbb{N}} \tau \cdot x_{i,\tau} \leq 1, \forall i \in \mathcal{A}. \tag{C.2}$$

In the above formulation, each variable $x_{i,\tau}$ denotes the fraction of time where arm $i$ is played under delay $\tau$ – namely, exactly $\tau$ time steps after it was played for the last time. In ($\mathbf{LP}_k$), constraint (C.1) follows by the fact that the total fraction of time where any action can be played (under any delay) cannot be more than $k$. Further, constraints (C.2) are valid constraints for the fractional plays of any individual arm under different delays.

We start by proving that the optimal solution of ($\mathbf{LP}_k$) is asymptotically an upper bound on the optimal average expected payoff:

**Lemma 3.1.** *For any instance of the $k$-RB problem, let $V^*$ be the optimal value of ($\mathbf{LP}_k$) and $\mathrm{OPT}(T)$ be the maximum payoff that can be collected in a time horizon of $T$ rounds. Then $T \cdot V^* \geq \mathrm{OPT}(T)$.*

Assuming knowledge of an upper bound $\tau^{\max} \geq \max_{i \in \mathcal{A}}\{\tau_i^{\max}\}$ on the maximum recovery time of any action, an optimal extreme point solution to ($\mathbf{LP}_k$) can be computed in polynomial time:

**Fact 3.2.** *There exists an optimal solution to ($\mathbf{LP}_k$) which is supported in $(x_{i,\tau})_{i \in \mathcal{A}, \tau \in [\tau_i^{\max}]}$. Further, given an upper bound $\tau^{\max}$ on the maximum recovery time of any action, an optimal extreme point solution to ($\mathbf{LP}_k$) can be computed in polynomial time.*

Let us introduce the notion of *supported* actions:

**Definition 3.3** (Supported actions)**.** *For any (potentially infeasible) solution $\mathbf{x}$ to ($\mathbf{LP}_k$), we say that an action $i \in \mathcal{A}$ is* supported *in $\mathbf{x}$, denoted by $i \in \mathcal{A}(\mathbf{x})$, if there exists $\tau \in \mathbb{N}$ such that $x_{i,\tau} > 0$.*

We introduce the following class of solutions:

**Definition 3.4** (Delay-feasible solutions)**.** *We say that a vector $\mathbf{x} = (x_{i,\tau})_{i \in \mathcal{A}, \tau \in \mathbb{N}}$ is* delay-feasible, *if for each supported action $i \in \mathcal{A}(\mathbf{x})$ there exists at most one non-zero variable $x_{i,\tau_i}$, and for this variable it holds $x_{i,\tau_i} = 1/\tau_i$. Further, we say that $\mathbf{x}$ is* almost-delay-feasible, *if it is delay-feasible, possibly with the exception of a single arm $\iota \in \mathcal{A}(\mathbf{x})$, for which there exist at most two non-zero variables $x_{\iota,\tau_{\iota,a}}, x_{\iota,\tau_{\iota,b}}$, such that $x_{\iota,\tau_{\iota,a}} < 1/\tau_{\iota,a}$ and $x_{\iota,\tau_{\iota,b}} < 1/\tau_{\iota,b}$. In the case of almost-delay-feasible solutions, we refer to $\iota$ as the* irregular *arm.*

---

[3]A function $f : 2^{[n]} \to \mathbb{R}$ is submodular, if for any $S, T \subseteq [n]$, it holds $f(S \cup T) + f(S \cap T) \leq f(S) + f(T)$.

We remark that delay-feasible solutions subsume by definition almost-delay-feasible solutions and that a (almost-)delay-feasible solution is not necessarily a feasible solution to ($\mathbf{LP}_k$).

As we prove in the next key-lemma, any extreme point solution of ($\mathbf{LP}_k$) is almost-delay-feasible:

**Lemma 3.5** (Sparsity pattern of extreme point solutions)**.** *Let* $\mathbf{x}$ *be any extreme point solution of* ($\mathbf{LP}_k$)*. Then,* $\mathbf{x}$ *is almost-delay-feasible.*

# 4 Improved Approximation Guarantees for Planning

For any instance of $k$-RB, our algorithm (see Algorithm 1 below), called "Randomize-Then-Interleave", constructs a feasible planning schedule without requiring knowledge of the time horizon. We remark that for the planning setting of the problem, we treat the payoff of any arm under any possible delay as deterministic.

The algorithm starts from computing an optimal extreme point solution $\mathbf{x}^*$ to ($\mathbf{LP}_k$), and then uses this solution to determine a *critical* delay $\tau_i^*$ for each supported arm $i \in \mathcal{A}(\mathbf{x}^*)$. By Lemma 3.5, we know that $\mathbf{x}^*$ is almost-delay-feasible, which means that – possibly with the exception of a single irregular arm $\iota$ – for every arm $i \in \mathcal{A}(\mathbf{x}^*) \setminus \{\iota\}$, there exists a unique $\tau_i$, such that $x_{i,\tau_i}^* = 1/\tau_i > 0$. Thus, Algorithm 1 sets this unique $\tau_i$ to be the critical delay of each arm $i \in \mathcal{A}(\mathbf{x}^*) \setminus \{\iota\}$. In the case where an irregular arm $\iota \in \mathcal{A}(\mathbf{x}^*)$ exists, then, by Definition 3.4, there exist at most two distinct $\tau_{\iota,a}, \tau_{\iota,b}$ with $x_{\iota,\tau_{\iota,a}}^* > 0$ and $x_{\iota,\tau_{\iota,b}}^* \geq 0$, in which case the critical delay $\tau_\iota^*$ of $\iota$ is set to $\tau_{\iota,a}$ or $\tau_{\iota,b}$, with marginal probability $\tau_{\iota,a} x_{\iota,\tau_{\iota,a}}^*$ and $\tau_{\iota,b} x_{\iota,\tau_{\iota,b}}^*$, respectively. Notice that by constraints (C.2) of ($\mathbf{LP}_k$), the above sampling process is well-defined, since $\tau_{\iota,a} x_{\iota,\tau_{\iota,a}}^* + \tau_{\iota,b} x_{\iota,\tau_{\iota,b}}^* \leq 1$. In the case where no $\tau_\iota^*$ is sampled, the irregular arm is removed from $\mathcal{A}(\mathbf{x}^*)$. After computing the critical delays, for each arm $i \in \mathcal{A}(\mathbf{x}^*)$ the algorithm draws a random *offset* $r_i$ independently and uniformly from $\{0, 1, \ldots, \tau_i^* - 1\}$.

---

**Algorithm 1:** Randomize-Then-Interleave

---

`/* Initialization phase                                                    */`

Compute an optimal extreme point solution $\mathbf{x}^*$ to ($\mathbf{LP}_k$).

Let $\iota \in \mathcal{A}$ be the irregular arm (if such an arm exists).

**for** *each arm* $i \in \mathcal{A}(\mathbf{x}^*) \setminus \{\iota\}$ **do**

    Let $\tau_i^*$ be the unique $\tau \in \mathbb{N}$ which satisfies $x_{i,\tau}^* = 1/\tau$.

**if** *an irregular arm* $\iota \in \mathcal{A}(\mathbf{x}^*)$ *exists* **then**

    Let $\tau_{\iota,a}, \tau_{\iota,b} \in \mathbb{N}$ be such that $x_{\iota,\tau_{\iota,a}}^* > 0$, $x_{\iota,\tau_{\iota,b}}^* \geq 0$, and $\tau_{\iota,a} \neq \tau_{\iota,b}$.

    Set $p_a \leftarrow \tau_{\iota,a} x_{\iota,\tau_{\iota,a}}^*$ and $p_b \leftarrow \tau_{\iota,b} x_{\iota,\tau_{\iota,b}}^*$.

    Sample $\tau_\iota^*$ from $\{\tau_{\iota,a}, \tau_{\iota,b}, \infty\}$ with marginals $p_a, p_b$, and $1 - p_a - p_b$, respectively.

    **if** $\tau_\iota^* = \infty$ **then**

        Remove $\iota$ from $\mathcal{A}(\mathbf{x}^*)$.

**for** *each arm* $i \in \mathcal{A}(\mathbf{x}^*)$ **do**

    Sample an *offset* $r_i$ uniformly at random from $\{0, 1, \ldots, \tau_i^* - 1\}$.

`/* Online phase                                                            */`

**for** $t = 1, 2, \ldots$ **do**

    Let $C_t \subseteq \mathcal{A}$ be the subset of *candidate* arms, defined as $C_t = \{i \in \mathcal{A}(\mathbf{x}^*) \mid t \bmod \tau_i^* \equiv r_i\}$.

    Play the maximum-payoff subset of $k$ arms in $C_t$, namely, $A_t = \underset{S \subseteq C_t, |S| \leq k}{\mathrm{argmax}} \sum_{i \in S} p_i(\sigma_{i,t})$,

    where $\sigma_{i,t}$ (*actual delay*) is the time passed since arm $i$ was last played before $t$.

---

After initialization, Algorithm 1 proceeds to its online phase where, at each round $t$, it first computes a subset of *candidate* arms $C_t$, namely, all arms $i \in \mathcal{A}(\mathbf{x}^*)$ which satisfy $t \bmod \tau_i^* \equiv r_i$. Then, the algorithm computes the maximum-payoff subset $A_t$ of at most $k$ candidate arms, computed using their actual delays (the number of rounds passed since last play), and plays these arms.

## 4.1 Approximation Analysis: Assuming Delay-Feasible LP Solutions

We now present an analysis of the approximation guarantee of Algorithm 1 for general $k$. In order to facilitate the presentation, we first focus on a special class of "easy" instances, where the optimal

extreme point solution $\mathbf{x}^*$ of ($\mathbf{LP}_k$) is delay-feasible (i.e., there is no irregular arm). Consequently, we show how this assumption can be relaxed at the expense of additional technical effort.

Recall that we denote by $\mathcal{A}(\mathbf{x}^*)$ the set of arms that are supported in the solution $\mathbf{x}^*$. Notice that, since $\mathbf{x}^*$ is delay-feasible by assumption, the only source of randomness in Algorithm 1 is due to the random offsets. Since each offset $r_i$ is sampled independently and uniformly from $\{0, 1, \ldots, \tau_i^* - 1\}$ and $i \in C_t$ if and only if $t \bmod \tau_i^* \equiv r_i$, the following fact is immediate:

**Fact 4.1.** *Let* $\{\tau_i^*\}_{i \in \mathcal{A}(\mathbf{x}^*)}$ *be the critical delays of the supported arms. At any round $t$, each arm $i \in \mathcal{A}(\mathbf{x}^*)$ belongs to $C_t$ independently with probability $1/\tau_i^*$.*

Let us focus on the expected payoff collected by Algorithm 1 at any round $t \geq \tau^{\max}$. By our simplifying assumption that $\mathbf{x}^*$ is delay-feasible, for each supported arm $i \in \mathcal{A}(\mathbf{x}^*)$, there is a unique critical delay $\tau_i^*$, such that $x_{i,\tau_i^*}^* = 1/\tau_i^*$ is the only non-zero variable of $\mathbf{x}^*$ corresponding to $i$.

We construct a vector $\mathbf{y} = [0, 1]^n$, such that $y_i = 1/\tau_i^*$ for each $i \in \mathcal{A}(\mathbf{x}^*)$, and $y_i = 0$, otherwise. Further, let us define a vector $\mathbf{w} \in [0, \infty)^n$ such that $w_i = p_i(\tau_i^*)$ for each $i \in \mathcal{A}(\mathbf{x}^*)$, and $w_i = 0$, otherwise. Recall that for any fixed round $t \geq \tau^{\max}$, by definition of $C_t$ and the construction of the offsets, the events $\{i \in C_t\}_{i \in \mathcal{A}}$ are independent. Hence, we can think of $C_t$ as a random set, where each arm $i \in \mathcal{A}$ belongs to $C_t$ independently with probability $y_i$, namely, $C_t \sim \mathcal{I}(\mathbf{y})$.

Let $\underline{A}_t = \underset{S \subseteq C_t, |S| \leq k}{\operatorname{argmax}} \sum_{i \in S} p_i(\tau_i^*)$ be a set of at most $k$ arms in $C_t$ which maximizes the total payoff measured with respect to the critical delays. By construction of $C_t$, the actual delay $\sigma_{i,t}$ of each $i \in \mathcal{A}(\mathbf{x}^*)$ is (deterministically) lower bounded by $\tau_i^*$, since any arm must be a candidate in order to be played (which happens exactly every $\tau_i^*$ rounds). Thus, for the expected collected payoff of any round $t \geq \tau^{\max}$ we have

$$\underset{C_t \sim \mathcal{I}(\mathbf{y})}{\mathbb{E}} \left[ \max_{S \subseteq C_t, |S| \leq k} \sum_{i \in S} p_i(\sigma_{i,t}) \right] \geq \underset{C_t \sim \mathcal{I}(\mathbf{y})}{\mathbb{E}} \left[ \max_{S \subseteq C_t, |S| \leq k} \sum_{i \in S} p_i(\tau_i^*) \right] = \underset{C_t \sim \mathcal{I}(\mathbf{y})}{\mathbb{E}} \left[ \mathbf{w}(\underline{A}_t) \right]. \quad (2)$$

Using Equation (1) in combination with Definitions 2.1 and 2.2, we get

$$\underset{C_t \sim \mathcal{I}(\mathbf{y})}{\mathbb{E}} \left[ \mathbf{w}(\underline{A}_t) \right] = \underset{C_t \sim \mathcal{I}(\mathbf{y})}{\mathbb{E}} \left[ f_{\mathbf{w},k}(C_t) \right] = F_{\mathbf{w},k}(\mathbf{y}) \geq \left( 1 - \frac{k^k}{e^k k!} \right) \cdot f_{\mathbf{w},k}^+(\mathbf{y}), \quad (3)$$

where the last inequality follows by Lemma 2.3.

The following technical lemma relates $f_{\mathbf{w},k}^+(\mathbf{y})$ with optimal value of ($\mathbf{LP}_k$):

**Lemma 4.2.** *For any vectors $\mathbf{w} \in [0, \infty)^n$ and $\mathbf{y} \in [0, 1]^n$, let $f_{\mathbf{w},k}^+(\mathbf{y})$ be the concave closure of the weighted rank function of the rank-$k$ uniform matroid with weights $\mathbf{w}$, evaluated at $\mathbf{y}$. Then, if $\mathbf{y}$ satisfies $\| \mathbf{y} \|_1 \leq k$, it is the case that $f_{\mathbf{w},k}^+(\mathbf{y}) \geq \sum_{i \in [n]} w_i y_i$.*

Let $V^*$ be the optimal value of ($\mathbf{LP}_k$). Since $\| \mathbf{y} \|_1 \leq k$, then by applying Lemma 4.2 with vectors $\mathbf{w}$ and $\mathbf{y}$ as constructed above, we have that

$$f_{\mathbf{w},k}^+(\mathbf{y}) \geq \sum_{i \in [n]} w_i \cdot y_i = \sum_{i \in \mathcal{A}(\mathbf{x}^*)} p_i(\tau_i^*) \cdot x_{i,\tau_i^*}^* = V^*. \quad (4)$$

By combining Equations (2) to (4) with Lemma 3.1, it follows that, for any round $t \geq \tau^{\max}$, the expected payoff collected by Algorithm 1 is at least $\left( 1 - k^k/e^k k! \right)$-times the average optimal payoff. By using linearity of expectations together with the fact that all payoffs are bounded in $[0, 1]$, we can prove a long-run $\left( 1 - k^k/e^k k! \right)$-approximation guarantee (in expectation) for instances where $\mathbf{x}^*$ is delay-feasible.

In the rest of this section, we show how the above analysis can be extended for the general case where $\mathbf{x}^*$ is almost-delay-feasible (i.e., an irregular arm exists).

## 4.2 Approximation Analysis: Handling the Irregular Arm

We now extend the analysis of Section 4.1 and relax the assumption that the optimal extreme point solution of ($\mathbf{LP}_k$) is delay-feasible.

Let $\mathbf{x}^*$ be an optimal solution to (LP$_k$) and let $\iota \in \mathcal{A}(\mathbf{x}^*)$ be the irregular arm with $\tau^*_{\iota,a}$ and $\tau^*_{\iota,b}$ being the two potential critical delays such that $x^*_{\iota,\tau_{\iota,a}}, x^*_{\iota,\tau_{\iota,b}} \geq 0$ (with at least one of $x^*_{\iota,\tau_{\iota,a}}$ and $x^*_{\iota,\tau_{\iota,b}}$ being strictly positive, by Definition 3.4). Instead of conditioning on the sampled critical delay of the irregular arm $\iota$, we construct two (fictitious) parallel copies $\iota_a$ and $\iota_b$ – one for each possible realization (and associated payoff). Let us define $\mathcal{A}' = \mathcal{A}(\mathbf{x}^*) \setminus \{\iota\} \cup \{\iota_a, \iota_b\}$ to be the set of supported arms of $\mathbf{x}^*$, where we replace the irregular arm $\iota$ with the two parallel copies $\iota_a$ and $\iota_b$.

In the next lemma, we characterize the marginal probability that any arm in $\mathcal{A}'$ is added to the set of candidate arms at any round. Notice that, as opposed to Fact 4.1, the randomness in this case is taken over both the sampling of a critical delay for the irregular arm as well as the random offsets.

**Lemma 4.3.** *For any arm $i \in \mathcal{A}'$ and at any round $t$, we have that $\mathbb{P}\left[i \in C_t \text{ and } \tau^*_i = \tau\right] = x^*_{i,\tau}$.*

Notice that, for the parallel copies of the irregular arm, the events $\{\iota_a \in C_t\}$ and $\{\iota_b \in C_t\}$ are now dependent and, in particular, *mutually exclusive* (since, eventually, at most one of them is realized). However, we are able to show that the expected payoff collected at any round $t \geq \tau^{\max}$ can only decrease if the two events were instead independent with the same marginal probabilities.

For convenience, we relabel the arms in $\mathcal{A}'$ in non-increasing order of $p_i(\tau^*_i)$ (breaking ties arbitrarily). Note that by exchanging $\iota$ with the two copies $\iota_a$ and $\iota_b$ the cardinality of $\mathcal{A}'$ can be at most $n + 1$.

Let us construct a $(n+1)$-dimensional vector $\mathbf{y} \in [0,1]^{n+1}$, such that $y_i = x^*_{i,\tau^*_i}$ for any $i \in \mathcal{A}'$, and $y_i = 0$, otherwise. Further, we construct a weight vector $\mathbf{w} \in [0,\infty)^{n+1}$, such that $w_i = p_i(\tau^*_i)$ for any $i \in \mathcal{A}'$, and $w_i = 0$, otherwise. Let $\mathcal{D}(\mathbf{y})$ be a distribution over subsets of $\mathcal{A}'$, where every arm $i \in \mathcal{A}' \setminus \{\iota_a, \iota_b\}$ (except for $\iota_a$ and $\iota_b$) is added to a set $S \sim \mathcal{D}(\mathbf{y})$, independently, with probability $x^*_{i,\tau^*_i}$. The arms $\iota_a$ and $\iota_b$ are also added to $S \sim \mathcal{D}(\mathbf{y})$ with marginal probabilities $x^*_{\iota_a,\tau^*_{\iota_a}}$ and $x^*_{\iota_b,\tau^*_{\iota_b}}$, respectively, yet not independently (since they are mutually exclusive). Therefore, by the above definitions and Lemma 4.3, the set of candidate arms at any fixed round $t \geq \tau^{\max}$ is distributed as $C_t \sim \mathcal{D}(\mathbf{y})$.

Let us fix any round $t \geq \tau^{\max}$. Similarly to Section 4.1, by using the fact that the actual delay $\sigma_{i,\tau}$ of each arm $i \in \mathcal{A}'$ is deterministically lower bounded by its critical delay $\tau^*_i$, we can lower bound the expected payoff collected by our algorithm at time $t$ as

$$\underset{C_t \sim \mathcal{D}(\mathbf{y})}{\mathbb{E}}\left[\max_{S \subseteq C_t, |S| \leq k} \sum_{i \in S} p_i(\sigma_{i,t})\right] \geq \underset{C_t \sim \mathcal{D}(\mathbf{y})}{\mathbb{E}}\left[\max_{S \subseteq C_t, |S| \leq k} \sum_{i \in S} p_i(\tau^*_i)\right] = \underset{C_t \sim \mathcal{D}(\mathbf{y})}{\mathbb{E}}\left[f_{\mathbf{w},k}(C_t)\right],$$
(5)

where $\mathcal{D}(\mathbf{y})$ is defined as described above.

As we show in the following lemma, the RHS of the above inequality can only decrease, if we replace $C_t \sim \mathcal{D}(\mathbf{y})$ with $C_t \sim \mathcal{I}(\mathbf{y})$, namely, if we assume that the two copies $\iota_a$ and $\iota_b$ are added to $C_t$ independently, but with the same marginals ($y_{\iota_a}$ and $y_{\iota_b}$, respectively). As we show in the following lemma, the above property in fact holds for any submodular function (including $f_{\mathbf{w},k}$):

**Lemma 4.4.** *Let $f : 2^{[n]} \to \mathbb{R}$ be a submodular function over a ground set of $n$ elements, where $n \geq 2$. Let $\mathcal{I}(\mathbf{y})$ be a distribution over $2^{[n]}$, where each element $i$ is added in set $S \sim \mathcal{I}(\mathbf{y})$, independently, with probability $y_i$. Further, let $\mathcal{D}(\mathbf{y})$ be a distribution over $2^{[n]}$, where each element $i \in [n] \setminus \{\iota_a, \iota_b\}$ is added to a set $S \sim \mathcal{D}(\mathbf{y})$, independently, with probability $y_i$, except for two distinct elements $\iota_a, \iota_b$, whose addition to $S$ is mutually exclusive but with the same marginals $y_{\iota_a}$ and $y_{\iota_b}$, respectively. In this case, we have*

$$\underset{S \sim \mathcal{D}(\mathbf{y})}{\mathbb{E}}\left[f(S)\right] \geq \underset{S \sim \mathcal{I}(\mathbf{y})}{\mathbb{E}}\left[f(S)\right].$$

By combining inequality (5) with Lemma 4.4 for $n + 1$ elements (recall, the function $f_{\mathbf{w},k}$ is submodular), the expected reward collected at any round $t \geq \tau^{\max}$ can be lower bounded as

$$\underset{C_t \sim \mathcal{D}(\mathbf{y})}{\mathbb{E}}\left[\max_{S \subseteq C_t, |S| \leq k} \sum_{i \in S} p_i(\sigma_{i,t})\right] \geq \underset{C_t \sim \mathcal{D}(\mathbf{y})}{\mathbb{E}}\left[f_{\mathbf{w},k}(C_t)\right] \geq \underset{C_t \sim \mathcal{I}(\mathbf{y})}{\mathbb{E}}\left[f_{\mathbf{w},k}(C_t)\right] = F_{\mathbf{w},k}(\mathbf{y}).$$

In addition, by combining Lemma 2.3 with Lemma 4.2 for the above choice of vectors $\mathbf{w}$ and $\mathbf{y}$ (again, it holds $\|\mathbf{y}\|_1 \leq k$), for any $t \geq \tau^{\max}$, it follows that

$$\mathbb{E}_{C_t \sim \mathcal{D}(\mathbf{y})} \left[ \max_{S \subseteq C_t, |S| \leq k} \sum_{i \in S} p_i(\sigma_{i,t}) \right] \geq F_{\mathbf{w},k}(\mathbf{y}) \geq \left( 1 - \frac{k^k}{e^k k!} \right) \cdot f_{\mathbf{w},k}^+(\mathbf{y}) \geq \left( 1 - \frac{k^k}{e^k k!} \right) V^*,$$

where $V^*$ is the optimal value of ($\mathbf{LP}_k$).

Finally, using the fact that $V^* \geq \mathrm{OPT}(T)/T$ by Lemma 3.1 and applying linearity of expectations over all rounds $t$, the following result follows directly:

**Theorem 4.5.** *For any instance of $k$-RB, where $\mathrm{OPT}(T)$ is the optimal expected payoff that can be collected in $T$ rounds, the total expected payoff collected by Algorithm 1 is at least*

$$\left( 1 - \frac{k^k}{e^k k!} \right) \mathrm{OPT}(T) - \mathcal{O}(k \cdot \tau^{\max}) \quad \approx \quad \left( 1 - \frac{1}{\sqrt{2\pi k}} \right) \mathrm{OPT}(T) - \mathcal{O}(k \cdot \tau^{\max}).$$

We remark that, although the asymptotic approximation guarantee of Algorithm 1 is the same as the one in [SLZZ21], the constants of our algorithm are significantly better. In particular, for $k \in \{1, 2, 3, 4, 5, 10\}$, the guarantees of [SLZZ21] and Algorithm 1 are $\{0.25, 0.33, 0.40, 0.44, 0.46, 0.57\}$ and $\{0.63, 0.72, 0.77, 0.80, 0.82, 0.87\}$, respectively.

# 5 Online Learning with Sublinear Regret

In this section, we consider the learning setting of $k$-RB where the payoff functions are initially unknown. At each round, the player plays a subset of at most $k$ arms and observes the realized payoff of each individual arm played (semi-bandit feedback). Every time an arm $i \in \mathcal{A}$ is played under delay $\tau$, the realized payoff is drawn independently from a distribution of mean $p_i(\tau)$, bounded in $[0, 1]$. For this setting, our goal is to show that there exists a bandit variant of Algorithm 1 with sublinear $\gamma_k$-approximate regret guarantee for $k$-RB, defined as

$$\mathrm{Reg}_{\gamma_k}(T) = \gamma_k \mathrm{OPT}(T) - \mathrm{R}(T),$$

where $\gamma_k = 1 - k^k/e^k k!$ is the (multiplicative) approximation guarantee of Algorithm 1 and $R(T)$ is the total expected payoff collected by the bandit algorithm in $T$ rounds.

**Robustness of Algorithm 1.** As a first step, we study the robustness of Algorithm 1 under small perturbations of the payoff functions; in particular, we analyze its approximation guarantee when it runs using estimates $\{\widehat{p}_i\}_{i \in \mathcal{A}}$ instead of the actual payoffs $\{p_i\}_{i \in \mathcal{A}}$. We remark that the monotonicity of the (estimated) payoff functions is *not required* for the correctness of Algorithm 1, but only for proving its approximation guarantee.

**Lemma 5.1.** *Let $\widehat{p}_i$ be an estimate of the payoff function $p_i$ for each $i \in \mathcal{A}$, such that $|\widehat{p}_i(\tau) - p_i(\tau)| \leq \epsilon, \forall \tau \in \mathbb{N}$ for some $\epsilon \in (0, 1)$. The expected payoff collected in $T$ rounds by Algorithm 1, when it uses the estimates instead of the actual payoff functions, is lower bounded by*

$$\gamma_k \mathrm{OPT}(T) - \mathcal{O}(k \cdot \epsilon \cdot T + k \cdot \tau^{\max}).$$

**Estimation of the payoff functions.** By using standard concentration results [Hoe63], we bound the number of samples required for each arm-delay pair in order to get an $\epsilon$-estimate $\widehat{p}_i$ for the mean payoff function $p_i$ of every arm $i \in \mathcal{A}$.

**Lemma 5.2.** *For any $\epsilon, \delta \in (0, 1)$, let $\widehat{p}_i(\tau)$ be the empirical average of samples drawn from $p_i(\tau)$ for every arm $i \in \mathcal{A}$ and delay $\tau$, using $m = \frac{1}{2\epsilon^2} \ln \left( \frac{2\tau^{\max}n}{\delta} \right)$ samples, where $n$ is the number of arms. Then, with probability $1 - \delta$, it holds $|\widehat{p}_i(\tau) - p_i(\tau)| \leq \epsilon$ for every $i \in \mathcal{A}$ and $\tau \in [\tau^{\max}]$.*

The following observation provides an upper bound on the number of rounds required in order to get $m$ samples for each arm-delay pair in any $k$-RB instance:

**Fact 5.3.** *In any $k$-RB instance, the number of rounds required to collect $m$ independent samples from $p_i(\tau)$, for each arm $i \in \mathcal{A}$ and delay $\tau \in [\tau^{\max}]$, is upper-bounded by $\mathcal{O}(nm(\tau^{\max})^2/k)$.*

**Bandit algorithm with sublinear regret.** By combining the above elements, it is easy to design an Explore-then-Commit (ETC) variant of Algorithm 1, achieving sublinear regret in the bandit setting. For simplicity, we assume here that the time horizon $T$ is known to the player – an assumption that can be relaxed using the doubling trick method [LS20].

Let $m = 1/2\epsilon^2 \ln(2\tau^{\max}n/\delta)$ for some $\epsilon, \delta \in (0,1)$. In the first $\mathcal{O}(nm(\tau^{\max})^2/k)$ rounds, the bandit algorithm collects $m$ independent samples from $p_i(\tau)$, for each $i \in \mathcal{A}$ and $\tau \in [\tau^{\max}]$. Then, by taking the empirical average of these samples, it constructs estimates $\widehat{p}_i(\tau)$ for each $i$ and $\tau$. For the rest of the time horizon, the algorithm simulates Algorithm 1, using the empirical estimates $\widehat{p}_i(\tau)$ in place of the (initially unknown) payoff functions.

By setting the parameters $\epsilon, \delta$ accordingly and combining Lemmas 5.1 and 5.2 with Fact 5.3, we prove the following result:

**Theorem 5.4.** *There exists a learning adaptation of Algorithm 1 in the semi-bandit setting, for which the $\gamma_k$-approximate regret for $T$ rounds can be upper-bounded as*

$$\mathcal{O}\left(n^{1/3} \cdot (k \cdot \tau^{\max})^{2/3} \ln^{1/3}(\tau^{\max}nT) \cdot T^{2/3} + k \cdot \tau^{\max}\right).$$

## Conclusion and Further Directions

Following the recent line of work on non-stationary bandits with recharging payoffs, we revisited one of the most general formulations studied in the area and significantly improved the state-of-the-art for the planning setting. In particular, when at most one arm is played per round, we designed an algorithm that collects at least $63\%$ (asymptotically and in expectation) of the optimal payoff, improving the best-known $25\%$, due to prior work. By providing results on the robustness and sample complexity of our algorithm, we transformed the latter into a bandit algorithm with sublinear regret. Our work leaves a number of interesting open questions. An approximation guarantee of $(1 - 1/e)$ has been proved to be the best achievable – under standard complexity assumptions – for a number combinatorial problems. Therefore, an immediate future direction would be to either provide a matching hardness result for the planning setting of the problem, or to further improve on the best-known approximation. Another interesting question would be to explore whether one can design a bandit adaptation of our algorithm with improved regret guarantees. Finally, the empirical evaluation of our algorithm on real (or artificial) data is another interesting direction.

## Acknowledgements

This work was partially funded by the NSF IFML Institute (NSF 2019844), the NSF AI-EDGE Institute (NSF 2112471), and NSF 1704778.

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
