# A  Related Work

Since the work of Thompson [Tho33] and later [LR+85], the (stochastic) multi-armed bandits (MAB) problem in stationary environments has been thoroughly studied (see [BCB12, LS20, ACBF02] and references therein). In an attempt to capture non-stationary environments, where the payoff distributions change over time, a number of generalizations of MAB – which are also special cases of reinforcement learning (RL) – have been proposed. Two particularly interesting classes are *rested* and *restless* bandits [TL12], where each action is associated with a state-machine and has a different payoff distribution depending on its current state. In rested bandits [Git79, TL12], the state of each action (hence, its payoff distribution) changes stochastically only when the arm is played, while in restless bandits [Whi88, TL12] the state changes at every round, independently of the player's actions. Optimal algorithms for rested and restless bandits essentially result from solving Bellman's equations [Ber00] – an approach which requires an exponentially large state space in the number of arms. In fact, Papadimitriou and Tsitsiklis [PT99] show that it is PSPACE-hard to optimally solve (or even approximate) restless bandits, even in the simpler case where state transitions are deterministic. One of the first attempts to provide constant approximation algorithms for special cases of restless bandits is due to Guha et al. [GMS10], who give a 2-approximation for a special case of restless bandits, which they call "monotone" bandits. Another interesting class of non-stationary bandits is that of *sleeping* bandits [KNMS10, SGV20], which generalizes the standard MAB setting, given that only an adversarially chosen subset of actions can be played at each round. Finally, we address the reader to [BGZ14, LCM17, KZ17, AGO19] for additional non-stationary bandit models.

Shifting our focus to the case of "recharging" (or "recovering") payoffs, Immorlica and Kleinberg [IK18] first consider the 1-RB problem, under the additional assumption that the payoff functions are weakly concave and weakly increasing over the whole time horizon. For this problem, the authors provide a PTAS that achieves a $(1 - \epsilon)$-approximation (asymptotically) to the optimal planning, and show how it can be adapted into a learning algorithm with sublinear regret guarantees. To achieve their approximation, they first compute an "optimal" playing frequency for each arm via a concave relaxation, and then construct an approximate planning schedule, in which the rate of playing each arm closely approximates this frequency. In this direction, they authors combine enumeration techniques with the novel technique of *interleaved rounding*. The main idea behind the latter is that each arm is first associated with a particular (randomly perturbed) sequence of real numbers and, then, the arms are played sequentially according to the order they appear on the real line.

Basu et al. [BSSS19] study a variant of stochastic multi-armed bandits where, after each play, an arm becomes unavailable for a known number of rounds (blocking time). As they show, for the case of one arm per round, a simple greedy heuristic, which plays, at each round, the arm of highest expected payoff among the non-blocked arms, yields (asymptotically) a $(1 - 1/e)$-approximation. In order to prove this guarantee, the authors compare the payoff collected by their algorithm with the optimal solution of an LP-relaxation, by carefully characterizing the latter via analyzing its optimality conditions. Although not stated explicitly in [BSSS19], the "hard" constraint of not pulling a blocked arm is equivalent to choosing the payoff function of each arm to be a Heaviside step function parameterized by its blocking time and scaled by its baseline mean payoff. In other words, the payoff drops to zero for a fixed number of rounds (which, in this case coincides with the recovery time), and then rises up to a baseline. Since pulling a zero-payoff arm can only harm the overall solution, the two problems are (without loss of generality) equivalent, and [BSSS19] becomes a special case of 1-RB. Notice, however, that the greedy approach of [BSSS19] fails to provide any non-trivial approximation guarantee for the 1-RB setting.

Shortly after its introduction, the model of Basu et al. [BSSS19] has been also studied in contextual [BPCS21], adversarial [BCMTT20], and combinatorial environments [APB+21, PC21]. In [PC21], Papadigenopoulos and Caramanis study the blocking setting in the regime where more than one arms can be played at each round, subject to matroid constraints. For the planning setting, they provide a $(1 - 1/e)$-approximation algorithm (asymptotically and in expectation) for any matroid. Their technique, called *interleaved scheduling* – which is similar in spirit with the *interleaved rounding* method of [IK18] – is essentially a time-correlated arm-sampling scheme, which provides the following two guarantees: (i) at any fixed round, each arm $i$ is sampled with probability $1/\tau_i$ (the maximum possible in any optimal solution), where $\tau_i$ is the blocking time of the arm. (ii) Every time an arm is sampled, its payoff must have returned to its baseline (i.e., is never blocked). Notice that the algorithm of [PC21], which essentially chooses a maximum-payoff independent

set among the sampled arms of each round, also provides an alternative method for reaching the $(1 - 1/e)$-approximation of [BSSS19] for the special case of one arm per round.

Cella and Cesa-Bianchi [CCB20] consider a similar model to 1-RB, where they assume that all the actions have identical payoff functions up to different scaling (i.e., baseline payoff), but different and unknown recovery times. For the planning problem, by focusing on a simple class of *periodic ranking policies*, the authors develop an algorithm, accompanied by a worst-case approximation guarantee that depends on characteristics of the input (payoff functions, recovery times, and baselines). Finally, other special cases and variations of the $k$-RB problem include payoff functions described by linear dynamics [LKKLM21], sampled by Gaussian processes [PBG19], and more [LCCBGB22].

Simchi-Levi et al. [SLZZ21] provide the first $\mathcal{O}(1)$-approximation the general case of the $k$-RB problem. Specifically, by focusing on a particular class of policies, which they call *purely periodic*, the authors provide a long-run $(1 - \mathcal{O}(1/\sqrt{k}))$-approximation for any instance of $k$-RB. A key-element in both Immorlica and Kleinberg [IK18] and, subsequently, Simchi-Levi et al. [SLZZ21] is the construction of a concave relaxation of the optimal solution, based on the critical notion of *rate of return*: a piecewise linear function, which denotes the maximum payoff that can be achieved asymptotically by playing an arm at most a specific fraction of rounds. The authors in [IK18] provide an FPTAS for computing an $(1 - \epsilon)$-approximate solution to their relaxation (for the case where $k = 1$), while, in [SLZZ21], such a solution is efficiently computed assuming finite recovery times.

# B  Structural Properties of a Natural LP Relaxation: Omitted Proofs

**Lemma 3.1.** *For any instance of the $k$-RB problem, let $V^*$ be the optimal value of $(\mathbf{LP}_k)$ and $\mathrm{OPT}(T)$ be the maximum payoff that can be collected in a time horizon of $T$ rounds. Then $T \cdot V^* \geq \mathrm{OPT}(T)$.*

*Proof.* Let $n_{i,\tau}$ be the number of times arm $i$ is played under delay $\tau \in \mathbb{N}$ in an optimal solution. By fixing $x_{i,\tau} = \frac{n_{i,\tau}}{T}$, we have that

$$\mathrm{OPT}(T) = \sum_{i \in \mathcal{A}} \sum_{\tau \in \mathbb{N}} p_i(\tau) \cdot n_{i,\tau} = T \cdot \sum_{i \in \mathcal{A}} \sum_{\tau \in \mathbb{N}} p_i(\tau) \cdot \frac{n_{i,\tau}}{T} = T \cdot \sum_{i \in \mathcal{A}} \sum_{\tau \in \mathbb{N}} p_i(\tau) \cdot x_{i,\tau}.$$

Further, note that in any instance where at most $k$ arms can be played at each round, the total number of arm plays (independently of the delay) cannot be more than $k \cdot T$. Thus, we get

$$\sum_{i \in \mathcal{A}} \sum_{\tau \in \mathbb{N}} x_{i,\tau} = \frac{1}{T} \cdot \sum_{i \in \mathcal{A}} \sum_{\tau \in \mathbb{N}} n_{i,\tau} \leq \frac{1}{T} \cdot k \cdot T \leq k.$$

Finally, consider any action $i \in \mathcal{A}$ which is played at time $t$ under delay $\tau$. The number of rounds that are occupied for the above specific play belong to the interval $I = \{t - \tau + 1, t\}$, with $|I| = \tau$ (note that for $\tau = 1$ only round $t$ is occupied). Recalling our assumption that all actions start with delay 1 at round $t = 1$, then for any such interval we have that $I \subseteq [T]$. Combining the above with the fact that, for any action $i \in \mathcal{A}$, these intervals cannot overlap, we get that

$$\sum_{\tau \in \mathbb{N}} \tau \cdot n_{i,\tau} \leq T \implies \sum_{\tau \in \mathbb{N}} \tau \cdot x_{i,\tau} \leq 1, \qquad \forall i \in \mathcal{A}.$$

Therefore, since $\mathbf{x}$ is a feasible solution to $(\mathbf{LP}_k)$ with value equal to $\mathrm{OPT}(T)/T$, the proof follows directly. $\qquad\square$

**Fact 3.2.** *There exists an optimal solution to $(\mathbf{LP}_k)$ which is supported in $(x_{i,\tau})_{i \in \mathcal{A}, \tau \in [\tau_i^{\max}]}$. Further, given an upper bound $\tau^{\max}$ on the maximum recovery time of any action, an optimal extreme point solution to $(\mathbf{LP}_k)$ can be computed in polynomial time.*

*Proof.* Clearly, proving that any optimal solution to $(\mathbf{LP}_k)$ is without loss of generality supported in $(x_{i,\tau})_{i \in \mathcal{A}, \tau \in [\tau_i^{\max}]}$ implies that we can restrict ourselves to a polynomial-size version of $(\mathbf{LP}_k)$, which is constructed by dropping all variables except for $(x_{i,\tau})_{i \in \mathcal{A}, \tau \in [\tau_i^{\max}]}$. In this way, an optimal extreme point solution to this restricted LP can be computed efficiently using a standard LP solver.

Let $\mathbf{x}$ be any optimal solution to $(\mathbf{LP}_k)$ such that there exists an arm $i$ and delay $\tau' > \tau_i^{\max}$ satisfying $x_{i,\tau'} > 0$. Note that by definition of the recovery time, it must hold $p_i(\tau') = p_i(\tau_i^{\max})$.

Thus, we can construct a solution $\mathbf{x}'$ which is identical to $\mathbf{x}$, except for the fact that $x'_{i,\tau'} = 0$ and $x'_{i,\tau_i^{\max}} = x_{i,\tau_i^{\max}} + x_{i,\tau'}$. Since $p_i(\tau') = p_i(\tau_i^{\max})$ the solutions $\mathbf{x}'$ and $\mathbf{x}$ have exactly the same value, while $\mathbf{x}'$ trivially satisfies constraints (C.1). In addition, constraints (C.2) are still satisfied by $\mathbf{x}'$ for any arm different than $i$. For arm $i$, we have

$$
\begin{aligned}
\sum_{\tau \in \mathbb{N}} \tau \cdot x'_{i,\tau} &= \sum_{\tau \in \mathbb{N} \setminus \{\tau', \tau_i^{\max}\}} \tau \cdot x_{i,\tau} + \tau_i^{\max} \cdot x'_{i,\tau_i^{\max}} \\
&= \sum_{\tau \in \mathbb{N} \setminus \{\tau', \tau_i^{\max}\}} \tau \cdot x_{i,\tau} + \tau_i^{\max} \cdot x_{i,\tau_i^{\max}} + \tau_i^{\max} \cdot x_{i,\tau'} \\
&\leq \sum_{\tau \in \mathbb{N}} \tau \cdot x_{i,\tau} \\
&\leq 1,
\end{aligned}
$$

where the first inequality follows by the fact that $\tau' > \tau_i^{\max}$ and the second by feasibility of $\mathbf{x}$.

By repeating the above process for every arm $i$ and delay $\tau > \tau_i^{\max}$ with $x_{i,\tau'} > 0$, we can transform any solution to $(\mathbf{LP}_k)$ to an equivalent solution of the desired form. $\square$

**Lemma 3.5** (Sparsity pattern of extreme point solutions). *Let $\mathbf{x}$ be any extreme point solution of $(\mathbf{LP}_k)$. Then, $\mathbf{x}$ is almost-delay-feasible.*

*Proof.* Let $\mathcal{A}(\mathbf{x}) \subseteq \mathcal{A}$ be the set of supported actions in an extreme point solution $\mathbf{x}$ of $(\mathbf{LP}_k)$, and let $|\mathcal{A}(\mathbf{x})| = n'$. Let LP$'$ be the LP that results by dropping from $(\mathbf{LP}_k)$ all the variables and constraints that correspond to the non-supported actions in $\mathcal{A} \setminus \mathcal{A}(\mathbf{x})$. Then, the solution $\mathbf{x}' = (x_{i,\tau})_{i \in \mathcal{A}(\mathbf{x}), \tau \in \mathbb{N}}$ (i.e., the solution $\mathbf{x}$ restricted to the arms of $\mathcal{A}(\mathbf{x})$) is an extreme point solution of this reduced LP. In order to see that, let us assume that $\mathbf{x}'$ is not an extreme point and, hence, $\mathbf{x}' = \lambda \mathbf{x}'_1 + (1 - \lambda) \mathbf{x}'_2$ for some $\lambda \in (0,1)$ and $\mathbf{x}'_1, \mathbf{x}'_2$ feasible solutions to LP$'$. Then, by padding $\mathbf{x}'_1$ and $\mathbf{x}'_2$ with zero variables for any $i \in \mathcal{A} \setminus \mathcal{A}(\mathbf{x})$ and $\tau \in \mathbb{N}$, the resulting vectors, let $\mathbf{x}_1$ and $\mathbf{x}_2$, respectively, are feasible solutions to $(\mathbf{LP}_k)$. However, it also holds that $\mathbf{x} = \lambda \mathbf{x}_1 + (1 - \lambda) \mathbf{x}_2$, contradicting the fact that $\mathbf{x}$ is an extreme point of $(\mathbf{LP}_k)$.

Now let us focus on the reduced formulation LP$'$. By construction, since the actions in $\mathcal{A}(\mathbf{x}')$ are all supported in $\mathbf{x}'$, it has to be that $\|\mathbf{x}'\|_0 \geq n'$. Let $u$ be the number of variables of LP$'$. By counting constraint (C.1), constraints (C.2) (one for each $i \in \mathcal{A}(\mathbf{x}')$), and $u$ non-negativity constraints, we can see that the number of constraints in LP$'$ is $u + n' + 1$. Hence, since $\mathbf{x}'$ is an extreme point solution of LP$'$, it must contain at most $n' + 1$ non-zero variables (since at least $u - n' - 1$ of the inequalities which are tight in $\mathbf{x}'$ must correspond to non-negativity constraints). This gives $\|\mathbf{x}'\|_0 \leq n' + 1$ and, thus, we can conclude that $\|\mathbf{x}'\|_0 \in \{n', n' + 1\}$.

We now distinguish between two cases:

(a) If $\|\mathbf{x}'\|_0 = n'$, then since every arm in the solution $\mathbf{x}'$ is supported and $\mathcal{A}(\mathbf{x}') = n'$, by pigeonhole principle each arm $i \in \mathcal{A}(\mathbf{x}')$ has to be supported by a unique non-zero variable $x'_{i,\tau_i} > 0$ for some $\tau_i \in \mathbb{N}$. Further, since exactly $n'$ out of the $u$ non-negativity constraints are not tight, then $n'$ out of the $n' + 1$ inequalities in (C.1) and (C.2) have to be met with equality. (i) In the case where these $n'$ constraints come from (C.2), then for each $i \in \mathcal{A}(\mathbf{x}')$, there exists a unique delay $\tau_i$ and variable $x'_{i,\tau_i}$, such that $x'_{i,\tau_i} = 1/\tau_i$, implying that $\mathbf{x}'$ and, thus, $\mathbf{x}$ is delay-feasible. (ii) In the case where only $n' - 1$ from the tight constraints in (C.1) and (C.2) come from (C.2), then for each corresponding arm $i \in \mathcal{A}(\mathbf{x}')$, there exists a unique delay $\tau_i$ and variable $x'_{i,\tau_i}$, such that $x'_{i,\tau_i} = 1/\tau_i$. Therefore, the arm $\iota \in \mathcal{A}(\mathbf{x}')$ which is associated with the non-tight constraint in (C.2) has to be an arm which is supported by a single variable $0 < x_{\iota,\tau_\iota} < 1/\tau_\iota$. In this case, $\mathbf{x}'$ and, thus, $\mathbf{x}$ is almost-delay-feasible, and $\iota$ corresponds to the irregular arm.

(b) In the case where $\|\mathbf{x}'\|_0 = n' + 1$, by a similar argument as above, each arm $i \in \mathcal{A}(\mathbf{x}')$ has to be supported by a unique non-zero variable $x'_{i,\tau_i} > 0$ for some $\tau_i \in \mathbb{N}$, with the exception of a single arm $\iota \in \mathcal{A}(\mathbf{x}')$, which is supported by two non-zero variables $x_{\iota,\tau_{\iota,a}}, x_{\iota,\tau_{\iota,b}} > 0$. Further, since $\|\mathbf{x}'\|_0 = n' + 1$, it has to be that all the constraints in (C.1) and (C.2) are met with equality. Therefore, for any arm $i \in \mathcal{A}(\mathbf{x}') \setminus \{\iota\}$, there exists a unique delay $\tau_i$ and variable $x'_{i,\tau_i}$, such that $x'_{i,\tau_i} = 1/\tau_i$. Again, the solution $\mathbf{x}$ of $(\mathbf{LP}_k)$ in this case is almost-delay-feasible, and $\iota$ corresponds to the irregular arm.

Since delay-feasible solutions subsume almost-delay-feasible by definition, it follows that, in any case, $\mathbf{x}$ is almost-delay-feasible. $\qquad\square$

## C   Improved Approximation Guarantees for Planning: Omitted Proofs

**Lemma 4.2.** *For any vectors $\mathbf{w} \in [0, \infty)^n$ and $\mathbf{y} \in [0, 1]^n$, let $f_{\mathbf{w},k}^+(\mathbf{y})$ be the concave closure of the weighted rank function of the rank-$k$ uniform matroid with weights $\mathbf{w}$, evaluated at $\mathbf{y}$. Then, if $\mathbf{y}$ satisfies $\| \mathbf{y} \|_1 \leq k$, it is the case that $f_{\mathbf{w},k}^+(\mathbf{y}) \geq \sum_{i \in [n]} w_i y_i$.*

*Proof.* Let us consider the intersection of the $n$-dimensional hypercube $[0, 1]^n$ with the halfspace $H = \{\mathbf{z} \in \mathbb{R}^n \mid \sum_{i \in [n]} z_i \leq k\}$. It is easy to verify that every vertex $\mathbf{v}$ of the resulting polytope lies in $\{0, 1\}^n$ and satisfies $\|\mathbf{v}\|_0 \leq k$. Then, since $\mathbf{y} \in [0, 1]^n \cap H$, by standard arguments in polyhedral combinatorics, $\mathbf{y}$ can be expressed as a convex combination of characteristic vectors of vertices of $[0, 1]^n \cap H$, each corresponding to a subset of $[n]$ of cardinality at most $k$. This convex combination induces a probability distribution $\mathcal{D}'(\mathbf{y})$ over subsets of cardinality at most $k$ with marginals $\mathbf{y}$. Therefore, by definition of $f_{\mathbf{w},k}^+$, we have

$$f_{\mathbf{w},k}^+(\mathbf{y}) = \sup_{\mathcal{D}(\mathbf{y})} \mathop{\mathbb{E}}_{S \sim \mathcal{D}(\mathbf{y})} [f_{\mathbf{w},k}(S)] \geq \mathop{\mathbb{E}}_{S \sim \mathcal{D}'(\mathbf{y})} [f_{\mathbf{w},k}(S)] = \mathop{\mathbb{E}}_{S \sim \mathcal{D}'(\mathbf{y})} [\mathbf{w}(S)],$$

where the last equality follows by the fact that $\mathcal{D}'(\mathbf{y})$ is only supported by sets of cardinality at most $k$.

Finally, using the fact that $\mathcal{D}'(\mathbf{y})$ respects the marginals $\mathbf{y}$, we can conclude

$$f_{\mathbf{w},k}^+(\mathbf{y}) \geq \mathop{\mathbb{E}}_{S \sim \mathcal{D}'(\mathbf{y})} [\mathbf{w}(S)] = \sum_{i \in [n]} w_i \mathop{\mathbb{P}}_{S \sim \mathcal{D}'(\mathbf{y})} [i \in S] = \sum_{i \in [n]} w_i y_i.$$

$\qquad\square$

**Lemma 4.3.** *For any arm $i \in \mathcal{A}'$ and at any round $t$, we have that $\mathbb{P}[i \in C_t \text{ and } \tau_i^* = \tau] = x_{i,\tau}^*$.*

*Proof.* Fix any arm $i \in \mathcal{A}'$ and round $t$. By the sampling process of critical delays using the solution $\mathbf{x}^*$, we have that $\mathbb{P}[\tau_i^* = \tau] = \tau \cdot x_{i,\tau}^*$. Notice that, when arm $i$ is not the irregular arm, by Definition 3.4 this probability is identically one. Since the offset $r_i$ is sampled uniformly at random from $\{0, 1, \ldots, \tau_i^* - 1\}$ and $i \in C_t$ if and only if $t \bmod \tau_i^* = r_i$, we get that $\mathbb{P}[i \in C_t \mid \tau_i^* = \tau] = 1/\tau$. Hence, we can conclude that

$$\mathbb{P}[i \in C_t \text{ and } \tau_i^* = \tau] = \mathbb{P}[\tau_i^* = \tau] \cdot \mathbb{P}[i \in C_t \mid \tau_i^* = \tau] = (\tau \cdot x_{i,\tau}^*) \cdot \frac{1}{\tau} = x_{i,\tau}^*.$$

$\qquad\square$

**Lemma 4.4.** *Let $f : 2^{[n]} \to \mathbb{R}$ be a submodular function over a ground set of $n$ elements, where $n \geq 2$. Let $\mathcal{I}(\mathbf{y})$ be a distribution over $2^{[n]}$, where each element $i$ is added in set $S \sim \mathcal{I}(\mathbf{y})$, independently, with probability $y_i$. Further, let $\mathcal{D}(\mathbf{y})$ be a distribution over $2^{[n]}$, where each element $i \in [n] \setminus \{\iota_a, \iota_b\}$ is added to a set $S \sim \mathcal{D}(\mathbf{y})$, independently, with probability $y_i$, except for two distinct elements $\iota_a, \iota_b$, whose addition to $S$ is mutually exclusive but with the same marginals $y_{\iota_a}$ and $y_{\iota_b}$, respectively. In this case, we have*

$$\mathop{\mathbb{E}}_{S \sim \mathcal{D}(\mathbf{y})} [f(S)] \geq \mathop{\mathbb{E}}_{S \sim \mathcal{I}(\mathbf{y})} [f(S)].$$

*Proof.* By definition of the distribution $\mathcal{D}(\mathbf{y})$, since the events $\{\iota_a \in S\}$ and $\{\iota_b \in S\}$ are mutually exclusive, we have that $\mathop{\mathbb{P}}_{S \sim \mathcal{D}(\mathbf{y})} [\{\iota_a \in S\} \cap \{\iota_b \notin S\}] = \mathop{\mathbb{P}}_{S \sim \mathcal{D}(\mathbf{y})} [\iota_a \in S] = y_{\iota_a}$ and $\mathop{\mathbb{P}}_{S \sim \mathcal{D}(\mathbf{y})} [\{\iota_a \notin S\} \cap \{\iota_b \in S\}] = \mathop{\mathbb{P}}_{S \sim \mathcal{D}(\mathbf{y})} [\iota_b \in S] = y_{\iota_b}$. Further, we have that $\mathop{\mathbb{P}}_{S \sim \mathcal{D}(\mathbf{y})} [\{\iota_a \notin S\} \cap \{\iota_b \notin S\}] = 1 - \mathop{\mathbb{P}}_{S \sim \mathcal{D}(\mathbf{y})} [\{\iota_a \in S\} \cup \{\iota_b \in S\}] = 1 - y_{\iota_a} - y_{\iota_b}$.

Using the above, we can rewrite the LHS of the desired inequality as

$$\underset{S \sim \mathcal{D}(\mathbf{y})}{\mathbb{E}} [f(S)] =$$

$$\sum_{S' \subseteq [n] \setminus \{\iota_a, \iota_b\}} \prod_{i \in S'} y_i \prod_{\substack{i \notin S' \\ i \neq \iota_a, \iota_b}} (1 - y_i) \Big( (1 - y_{\iota_a} - y_{\iota_b}) f(S') + y_{\iota_a} f(S' + \iota_a) + y_{\iota_b} f(S' + \iota_b) \Big).$$

Now, for any fixed set $S' \subseteq [n] \setminus \{\iota_a, \iota_b\}$, the following identities can be easily verified:

$$(1 - y_{\iota_a} - y_{\iota_b}) f(S') = (1 - y_{\iota_a})(1 - y_{\iota_b}) f(S') - y_{\iota_a} y_{\iota_b} f(S')$$

$$y_{\iota_a} f(S' + \iota_a) = y_{\iota_a}(1 - y_{\iota_b}) f(S' + \iota_a) + y_{\iota_a} y_{\iota_b} f(S' + \iota_a)$$

$$y_{\iota_b} f(S' + \iota_b) = y_{\iota_b}(1 - y_{\iota_a}) f(S' + \iota_b) + y_{\iota_a} y_{\iota_b} f(S' + \iota_b).$$

By combining the above identities, we get that

$$(1 - y_{\iota_a} - y_{\iota_b}) f(S') + y_{\iota_a} f(S' + \iota_a) + y_{\iota_b} f(S' + \iota_b)$$

$$= (1 - y_{\iota_a})(1 - y_{\iota_b}) f(S') + y_{\iota_a}(1 - y_{\iota_b}) f(S' + \iota_a) + y_{\iota_b}(1 - y_{\iota_a}) f(S' + \iota_b)$$

$$+ y_{\iota_a} y_{\iota_b} \left( f(S' + \iota_a) + f(S' + \iota_b) - f(S') \right)$$

$$\geq (1 - y_{\iota_a})(1 - y_{\iota_b}) f(S') + y_{\iota_a}(1 - y_{\iota_b}) f(S' + \iota_a) + y_{\iota_b}(1 - y_{\iota_a}) f(S' + \iota_b) + y_{\iota_a} y_{\iota_b} f(S' + \iota_a + \iota_b),$$

where, in the inequality, we use that $f(S') + f(S' + \iota_a + \iota_b) \leq f(S' + \iota_a) + f(S' + \iota_b)$, by submodularity of $f$.

By applying the above inequality for each $S' \subseteq [n] \setminus \{\iota_a, \iota_b\}$, we can conclude that

$$\underset{S \sim \mathcal{D}(\mathbf{y})}{\mathbb{E}} [f(S)] = \sum_{S' \subseteq [n] \setminus \{\iota_a, \iota_b\}} \prod_{i \in S'} y_i \prod_{\substack{i \notin S' \\ i \neq \iota_a, \iota_b}} (1 - y_i) \Big( (1 - y_{\iota_a} - y_{\iota_b}) f(S') + y_{\iota_a} f(S' + \iota_a) + y_{\iota_b} f(S' + \iota_b) \Big)$$

$$\geq \sum_{S' \subseteq [n] \setminus \{\iota_a, \iota_b\}} \prod_{i \in S'} y_i \prod_{\substack{i \notin S' \\ i \neq \iota_a, \iota_b}} (1 - y_i) \Big( (1 - y_{\iota_a})(1 - y_{\iota_b}) f(S') + y_{\iota_a}(1 - y_{\iota_b}) f(S' + \iota_a)$$

$$+ y_{\iota_b}(1 - y_{\iota_a}) f(S' + \iota_b) + y_{\iota_a} y_{\iota_b} f(S' + \iota_a + \iota_b) \Big)$$

$$= \sum_{S \subseteq [n]} \prod_{i \in S} y_i \prod_{i \notin S} (1 - y_i) f(S)$$

$$= \underset{S \sim \mathcal{I}(\mathbf{y})}{\mathbb{E}} [f(S)],$$

thus proving the lemma. $\qquad\square$

## D  Online Learning with Sublinear Regret: Omitted Proofs

**Lemma 5.1.** *Let $\widehat{p}_i$ be an estimate of the payoff function $p_i$ for each $i \in \mathcal{A}$, such that $|\widehat{p}_i(\tau) - p_i(\tau)| \leq \epsilon, \forall \tau \in \mathbb{N}$ for some $\epsilon \in (0, 1)$. The expected payoff collected in $T$ rounds by Algorithm 1, when it uses the estimates instead of the actual payoff functions, is lower bounded by*

$$\gamma_k \, \mathrm{OPT}(T) - \mathcal{O}(k \cdot \epsilon \cdot T + k \cdot \tau^{\max}).$$

*Proof.* Let $\mathbf{x}^*$ and $\widehat{\mathbf{x}}$ be the optimal solution of $(\mathbf{LP}_k)$ in the case where the payoff functions in the objective are given by $\{p_i\}_{i \in \mathcal{A}}$ and $\{\widehat{p}_i\}_{i \in \mathcal{A}}$, respectively. Let also $V^*$ and $\widehat{V}$ be the optimal value of $(\mathbf{LP}_k)$ in each of the aforementioned cases. We have

$$\widehat{V} = \sum_{i \in \mathcal{A}} \sum_{\tau \in \mathbb{N}} \widehat{p}_i(\tau) \cdot \widehat{x}_{i,\tau} \geq \sum_{i \in \mathcal{A}} \sum_{\tau \in \mathbb{N}} \widehat{p}_i(\tau) \cdot x^*_{i,\tau} \geq \sum_{i \in \mathcal{A}} \sum_{\tau \in \mathbb{N}} (p_i(\tau) - \epsilon) \cdot x^*_{i,\tau} \geq V^* - \epsilon \cdot k,$$

where in the first inequality we use the fact that $\widehat{\mathbf{x}}$ is an optimal extreme point solution of $(\mathbf{LP}_k)$, when the estimates are used in the objective. In the second inequality, we use that $|\widehat{p}_i(\tau) - p_i(\tau)| \leq \epsilon, \forall i \in \mathcal{A}, \tau \in \mathbb{N}$, while the third follows by constraint (C.1) of $(\mathbf{LP}_k)$.

We observe that Algorithm 1 can produce a feasible planning using any given extreme point solution of ($\mathbf{LP}_k$) (not necessarily the optimal). Let $\widehat{A}_t$ be the set of arms played at some round $t \geq \tau^{\max}$ by Algorithm 1 (using the estimates) and let $\widehat{C}_t$ be the corresponding set of candidate arms. By construction of the algorithm, it is the case that

$$\widehat{A}_t = \operatorname*{argmax}_{S \subseteq \widehat{C}_t, |S| \leq k} \sum_{i \in S} \widehat{p}_i(\widehat{\sigma}_{i,t}),$$

where $\widehat{\sigma}_{i,t}$ is the actual delay of arm $i$ at time $t$ in a run of Algorithm 1 (using the estimates).

Let $\widehat{\tau}_i^*$ be the (sampled) critical delay of arm $i$ in a run of Algorithm 1 (using the estimates). By working along the lines of Theorem 4.5, using monotonicity of $p_i(\tau)$ and the fact that $\widehat{\sigma}_{i,t} \geq \widehat{\tau}_i^*$, for the expected payoff collected at any round $t \geq \tau^{\max}$ (over the randomness of the payoff realizations), we have

$$\max_{S \subseteq \widehat{C}_t, |S| \leq k} \sum_{i \in S} p_i(\widehat{\sigma}_{i,t}) \geq \max_{S \subseteq \widehat{C}_t, |S| \leq k} \sum_{i \in S} p_i(\widehat{\tau}_i^*) \geq \max_{S \subseteq \widehat{C}_t, |S| \leq k} \sum_{i \in S} \widehat{p}_i(\widehat{\tau}_i^*) - \epsilon \cdot k,$$

where in the last inequality we use the fact that $\widehat{p}_i$ is $\epsilon$-close to $p_i$ for each arm $i \in \mathcal{A}$.

Let $\gamma_k$ be the multiplicative approximation ratio of Algorithm 1 for $k$-RB (given in Theorem 4.5). By the analysis of Theorem 4.5, it follows that

$$\mathbb{E}\left[\max_{S \subseteq \widehat{C}_t, |S| \leq k} \sum_{i \in S} \widehat{p}_i(\widehat{\tau}_i^*)\right] \geq \gamma_k \cdot \widehat{V} \geq \gamma_k \cdot V^* - \gamma_k \cdot \epsilon \cdot k \geq \gamma_k \cdot \frac{\mathrm{OPT}(T)}{T} - \gamma_k \cdot \epsilon \cdot k,$$

where the last inequality follows by Lemma 3.1.

Therefore, for any round $t \geq \tau^{\max}$, the expected payoff collected by Algorithm 1 (using the estimates) satisfies

$$\mathbb{E}\left[\max_{S \subseteq \widehat{C}_t, |S| \leq k} \sum_{i \in S} p_i(\widehat{\sigma}_{i,t})\right] \geq \gamma_k \cdot \frac{\mathrm{OPT}(T)}{T} - \mathcal{O}(\epsilon \cdot k).$$

The proof follows by summing the above inequalities for every round $t \in [\tau^{\max}, T]$ and using linearity of expectation together with the fact that $\mathrm{OPT}(T)/T \leq k$. $\qquad\square$

**Lemma 5.2.** *For any $\epsilon, \delta \in (0, 1)$, let $\widehat{p}_i(\tau)$ be the empirical average of samples drawn from $p_i(\tau)$ for every arm $i \in \mathcal{A}$ and delay $\tau$, using $m = \frac{1}{2\epsilon^2} \ln\left(\frac{2\tau^{\max}n}{\delta}\right)$ samples, where $n$ is the number of arms. Then, with probability $1 - \delta$, it holds $|\widehat{p}_i(\tau) - p_i(\tau)| \leq \epsilon$ for every $i \in \mathcal{A}$ and $\tau \in [\tau^{\max}]$.*

*Proof.* For any arm $i \in \mathcal{A}$ and delay $\tau$, let $\widehat{p}_i(\tau)$ be the empirical average of $m$ samples drawn independently from a distribution of mean $p_i(\tau)$ (and bounded in $[0, 1]$). By Hoeffding inequality [Hoe63], we have that

$$\mathbb{P}\left[|\widehat{p}_i(\tau) - p_i(\tau)| > \epsilon\right] \leq 2\exp\left(-2m\epsilon^2\right).$$

By setting $m = \frac{1}{2\epsilon^2} \ln\left(\frac{2\tau^{\max}n}{\delta}\right)$, we get that $|\widehat{p}_i(\tau) - p_i(\tau)| \leq \epsilon$ with probability $1 - \frac{\delta}{\tau^{\max}n}$.

Therefore, by a simple union bound over all arms $i \in \mathcal{A}$ and delays $\tau \in [\tau^{\max}]$, we get

$$\mathbb{P}\left[\exists i \in \mathcal{A}, \tau \in [\tau^{\max}] : |\widehat{p}_i(\tau) - p_i(\tau)| > \epsilon\right] \leq \delta,$$

thus, conlcuding the proof. $\qquad\square$

**Fact 5.3.** *In any $k$-RB instance, the number of rounds required to collect $m$ independent samples from $p_i(\tau)$, for each arm $i \in \mathcal{A}$ and delay $\tau \in [\tau^{\max}]$, is upper-bounded by $\mathcal{O}\left(nm(\tau^{\max})^2/k\right)$.*

*Proof.* For any arm $i \in \mathcal{A}$ and delay $\tau$, we can collect $m$ independent samples from the distribution of mean $p_i(\tau)$, by playing arm $i$ for $m$ times – once every $\tau$ rounds – which can be achieved in $\mathcal{O}(m\tau)$ time steps. In order to collect $m$ samples for any delay $\tau \in [\tau^{\max}]$ for a fixed arm $i$, we thus need $\mathcal{O}\left(\sum_{\tau \in [\tau^{\max}]} m\tau\right) = \mathcal{O}\left(m(\tau^{\max})^2\right)$ rounds in the worst case.

Assuming that we can play at most one arm per round, in order to collect $m$ samples for each arm/delay pair, we thus need $\mathcal{O}(nm(\tau^{\max})^2)$ rounds in total. In a $k$-RB instance, where we can play at most $k < n$ arms per round, we can parallelize and, thus, expedite the above sampling process by a factor of $k$, leading to a total of $\mathcal{O}(nm(\tau^{\max})^2/k)$ exploration rounds. $\qquad\square$

**Theorem 5.4.** *There exists a learning adaptation of Algorithm 1 in the semi-bandit setting, for which the $\gamma_k$-approximate regret for $T$ rounds can be upper-bounded as*

$$\mathcal{O}\left(n^{1/3} \cdot (k \cdot \tau^{\max})^{2/3} \ln^{1/3}(\tau^{\max} nT) \cdot T^{2/3} + k \cdot \tau^{\max}\right).$$

*Proof.* Let $\mathrm{R}(T)$ be the expected payoff collected by our ETC-based bandit algorithm in $T$ rounds. Recall that the $\gamma_k$-approximate regret for $T$ rounds is defined as

$$\mathrm{Reg}_{\gamma_k}(T) = \gamma_k \,\mathrm{OPT}(T) - \mathrm{R}(T).$$

For $\epsilon, \delta \in (0,1)$ (to be specified later), let $m = \frac{1}{2\epsilon^2} \ln\left(2\tau^{\max} n/\delta\right)$. By Fact 5.3, we can collect $m$ samples from the distribution of each arm-delay pair in the first $\mathcal{O}\left(\frac{n(\tau^{\max})^2}{\epsilon^2 k} \ln(\frac{\tau^{\max} n}{\delta})\right)$ rounds. Since the mean payoffs are all bounded in $[0,1]$ and we can play at most $k$ arms per round, the regret accumulated for each of the exploration rounds is at most $k$, which implies that the total regret due to exploration can be upper bounded by $\mathcal{O}\left(\frac{n(\tau^{\max})^2}{\epsilon^2} \ln(\frac{\tau^{\max} n}{\delta})\right)$.

Moving on to the exploitation phase, for every arm $i \in \mathcal{A}$ and delay $\tau \in [\tau^{\max}]$, let $\widehat{p}_i(\tau)$ be the empirical estimate of $p_i(\tau)$ using $m$ samples collected in the exploration phase. For the rest $T - \mathcal{O}(\frac{n(\tau^{\max})^2}{\epsilon^2 k} \ln(\frac{\tau^{\max} n}{\delta})) \leq T$ rounds, by Lemma 5.2, we get that, with probability $1 - \delta$, it holds $|\widehat{p}_i(\tau) - p_i(\tau)| \leq \epsilon$ for every arm $i$ and delay $\tau$. We refer to the above event as a "nice" sampling.

Thus, the total regret accumulated in the exploitation phase in the case where sampling is not nice (which happens with probability $\delta$) is at most $T \cdot k \cdot \delta$. In the case where the sampling is nice, then Lemma 5.1 suggests that for the remaining $T'$ rounds of the exploitation phase, we have

$$\mathrm{R}(T') \geq \gamma_k \,\mathrm{OPT}(T') - \mathcal{O}(k \cdot \epsilon \cdot T' + k \cdot \tau^{\max}),$$

which implies that the total regret accumulated in the exploitation phase can be upper bounded by $\mathcal{O}(k \cdot \epsilon \cdot T + k \cdot \tau^{\max} + T \cdot k \cdot \delta)$. By the above analysis, it follows that for any $\epsilon, \delta \in (0,1)$, the total regret of our bandit algorithm satisfies

$$\mathrm{Reg}_{\gamma_k}(T) \leq \underbrace{\mathcal{O}\left(\frac{n(\tau^{\max})^2}{\epsilon^2} \ln(\frac{\tau^{\max} n}{\delta})\right)}_{\text{Exploration phase}} + \underbrace{\mathcal{O}\left(k \cdot \epsilon \cdot T + k \cdot \tau^{\max} + T \cdot k \cdot \delta\right)}_{\text{Exploitation phase}}.$$

By setting $\epsilon = \mathcal{O}\left(\sqrt[3]{\frac{n(\tau^{\max})^2 \ln(\tau^{\max} nT)}{kT}}\right)$ and $\delta = \frac{1}{T}$, we can upper bound the regret as

$$\mathrm{Reg}_{\gamma_k}(T) \leq \mathcal{O}\left(n^{1/3} \cdot (k \cdot \tau^{\max})^{2/3} \ln^{1/3}(\tau^{\max} nT) \cdot T^{2/3} + k \cdot \tau^{\max}\right),$$

thus proving that our algorithm achieves a sublinear (in the time horizon) regret guarantee. $\qquad \square$

## Societal Impact and Limitations

This is a theoretical work and, thus, any (positive or negative) societal impact depends on the application. A possible limitation of our work is the assumption that an upper bound on the recovery times of all arms is known a priori. Notice, however, this assumption (also made in the state-of-the-art prior work [SLZZ21]) is natural for settings where the payoff functions have a non-parametric form.