# OpenReview forum: "Non-Stationary Bandits under Recharging Payoffs: Improved Planning with Sublinear Regret"
_NeurIPS.cc/2022/Conference — NeurIPS 2022 Accept_

### Official Review · Reviewer_3yso · 2022-07-08

**Rating:** 7
**Confidence:** 5
**Soundness:** 3 good
**Presentation:** 3 good
**Contribution:** 3 good

**Summary:**

Authors analyze and improved the existing results for the recharging bandits problem. They put a lot of efforts in the optimization/planning part and obtained strong improvements on that side.

**Questions:**

I have only few questions.

The first question I have is which technical difficulties authors have met in improving the planning stage. Specifically, I would like to understand how much the results of [PC21] and [Yan11] contribute in the obtained improvements.

Secondly, I am wondering why developing an ETC approach rather than an UCB one as done by [Kleinberg and Immorlica].

**Limitations:**

See above.

**Strengths And Weaknesses:**

The paper is well written and the contributions are clearly stated in particular thanks to paragraph 1.3. I found the literature review to be very complete.

---

> ### Author Response · Authors · 2022-08-02
> **Response to Reviewer 3yso**
>
> We would like to thank the reviewer for the valuable feedback.
>
> The main technical difficulties we faced in improving the planning stage can be summarized as follows:
>
> * **Proving that any extreme point solution of our LP relaxation is sparse and, in particular, almost-delay-feasible** (see Lemma 3.5). This result alone already implies that any instance of recharging bandits (almost) corresponds to a simpler proxy instance with Heaviside step payoff functions.
> * **Extending the analysis of [PC21]**. The method of [PC21] yields a (1-1\e)-approximation for the case where the payoff functions are Heaviside step functions. By leveraging the results of [Yan11] on k-uniform combinatorial auctions, we improve the analysis of [PC21] and, thus, recover the desired scaling of 1-1/O(\sqrt(k)) for this particular setting, where at most k arms can be pulled at each round. However, the fact that the solution of our LP cannot be readily transformed into a Heaviside instance required additional technical contributions:
>      * **Handling the presence of an irregular arm**. Notice that the algorithm of [PC21] for the case of Heaviside step payoffs can be only applied if a unique integer blocking time is known for each arm. Via our randomized rounding scheme for sampling a critical delay for the irregular arm, we are able to construct such an instance (with integer blocking times), while at the same time maintaining the marginal contribution of each arm-delay pair, as it appears in the LP (see Lemma 4.4).
>      * **Correlation gap for non independent elements**. The original results of  [Yan11] on the correlation gap (see Lemma 2.3 in our paper) require that a random set is constructed by adding each element independently with some known probability. However, due to the existence of an irregular arm – which can be associated with two different critical delays – the set of “candidate” arms of each round is not constructed by adding each element independently. As we show in Lemma 4.5, this is not an issue for k=1, since the two different (mutually exclusive) realizations of the critical delay of the irregular arm can be treated as independent (with the same marginals). Finally, for k>1, we show that this issue does not affect the asymptotic guarantees of our algorithm (see Appendix C).
>
> The main focus of our work is the design of an algorithm with improved approximation guarantees for the planning problem. By developing an ETC approach, we show how our algorithm can be easily adapted into an algorithm of sublinear regret.
> Indeed, by developing a phase-based UCB approach, as done by [Kleinberg and Immorlica], it might be possible to achieve improved regret guarantees. However, achieving a smaller and smoother regret in our setting has a number of extra complications, compared to [Kleinberg and Immorlica]. For example, apart from the fact that we do not assume knowledge of the time horizon and can play more than one arms per round, it is critical that we cannot use extra computational effort to compensate for the hardness of obtaining diverse samples, due to the complicated nature of our algorithm. Notice that the latter is possible in [Kleinberg and Immorlica], by controlling the parameter \epsilon in the approximation guarantee of their PTAS. We believe that overcoming these challenges and designing a –purely online– UCB algorithm with small regret is an interesting and challenging future direction.

---

### Official Review · Reviewer_Y2yC · 2022-07-11

**Rating:** 7
**Confidence:** 3
**Soundness:** 3 good
**Presentation:** 3 good
**Contribution:** 3 good

**Summary:**

The paper studies a non-stationary bandit problem with $n$ arms, where the mean reward of an arm is a non-decreasing function of its "idle time" since last pull. It is assumed that each arm fully "recovers" after being idle for a certain number of periods, i.e., its mean reward does not change any further. An upper bound on the recovery time of all $n$ arms is also assumed known. At any time, the decision maker can play a subset of at most $k < n$ arms upon which she collects the sum of their associated payoffs. As is the norm, the goal is to maximize the cumulative expected payoffs over $T$ rounds, with $T$ possibly being unknown a priori.

**Questions:**

Just out of curiosity: If T is also known, can the approximation ratio be improved? Alternatively, if T is known but not the max recovery time, can something be claimed at least for "large" T perhaps by using some appropriate sub-linear function thereof as a proxy for the latter?

**Limitations:**

This appears to have been well-addressed.

**Strengths And Weaknesses:**

It is well-known that computing an optimal policy even for the "planning" problem (assuming perfect knowledge of the mean reward functions) is NP-hard. Notably, the authors improve upon the current best known $1/4$-approximation guarantee for $k=1$ to $1-1/e$ achievable in polynomial time. For $k>1$ also, their algorithm achieves improved numerical guarantees vis-\`a-vis extant work. Lastly, a learning version is also considered where the mean reward functions are initially unknown and decision maker only observes noisy semi-bandit feedback for the collection of arms played. For this setting, the authors propose an algorithm based on the Explore-then-Commit principle that is shown to incur sub-linear regret relative to its "full information" planning counterpart.

Overall, the analysis appears sound (though I did not delve into details contained in the supplementary sections) and I reckon this paper makes solid theoretical contributions. I would therefore vote for an accept.

---

> ### Author Response · Authors · 2022-08-02
> **Response to Reviewer Y2yC**
>
> We would like to thank the reviewer for the valuable feedback.
> Whether the knowledge of T can be used to improve the approximation guarantee is a very good question. The fact that we do not require knowledge of the time horizon T is a byproduct of our algorithm; we do not see how knowledge of T can be used to improve the approximation guarantee.
>
> Our algorithm essentially works assuming knowledge of any upper bound on the maximum recovery time over all arms (not necessarily a tight one). However, since the algorithm’s running time depends on this upper bound (through the number of variables in the LP), we require that the latter is polynomial (in the size of the input) in order for our algorithm to be efficient. Hence, using any (sublinear) function of T as a proxy for the maximum recovery time cannot really work in a large horizon regime.

---

### Official Review · Reviewer_uVwc · 2022-07-12

**Rating:** 6
**Confidence:** 2
**Soundness:** 3 good
**Presentation:** 1 poor
**Contribution:** 3 good

**Summary:**

The author studies a problem in which the reward of arms is increasing as the arms are not selected in the last rounds. At first, the author studied the problem of solving such an issue when the parameters of the problem are known. The first issue is to design an algorithm that effectively approximates the optimal solution to this problem. The author provide such an algorithm and studied its approximation properties. Finally, it provides an ETC scheme to deal with the case in which the learner is acquiring only bandit feedback. Finally, guarantees in terms of regret for this algorithm are provided.

**Questions:**


Why are the proof so important? Are they providing novel techniques which are of general interest?

**Limitations:**

I do not foresee limitations and potential negative societal impact of the current work.

**Strengths And Weaknesses:**

I think that the paper is well written. However, I think that the choice of the authors to include proofs and sketches of the prooofs is not effective to communicate the message of the paper. I think that such aspects, as well as the statement of some lemmas which are not central to the paper itself might have moved to the appendix to make room for some more comments and, possibly, to some final experiments.

I think that adding a table to summarize the different results present in the literature and the ones provided in this paper would help to appreciate the value of this work.

I think that adding a more formal definition of the bandit setting is necessary.

---

> ### Author Response · Authors · 2022-08-02
> **Response to Reviewer uVwc**
>
> We would like to thank the reviewer for the valuable feedback, especially concerning readability and presentation. We agree with many of the comments. Space constraints are tight, but the comments made us re-think some of the balancing we did. Accordingly:
>
> * In the past NeurIPS conferences, an additional page has been provided for the camera-ready version for addressing reviewer comments. If our paper is accepted and NeurIPS again provides this extra page, we can use this extra space to accommodate a table/summary comparing existing results in the literature with ours.
> * Similarly, the above extra space will be also used to present a more formal definition of the bandit setting.
> * We appreciate the reviewer’s comment regarding the valuable space taken up by technical lemmas and proof sketches. We will reconsider our approach; though we are trying to balance with the fact that, in our case, these elements play a central role in communicating our novel technical contributions and the main messages of our work. For instance, through our proofs we show that the general problem of planning recharging bandits can be reduced to the simpler special case of Heaviside step payoff functions (known as ``blocking bandits''). This implies that in order to improve (even more) the approximation guarantees for planning, one can focus on the significantly simpler case of Heaviside step payoff functions. We emphasize that the aforementioned property does not follow from the design of our algorithm, but from the structural analysis of the LP relaxation in Section 3.

---

### Meta-Review · Area_Chair_qQmj · 2022-09-11

**Recommendation:** Accept
**Confidence:** Certain

**Metareview:**

All reviewers agree on the merit of the work.

**Award:**

No

---

### Decision · Program_Chairs · 2022-09-14

Accept